# Investigation of the Reaction between a Homemade PEEK Oligomer and an Epoxy Prepolymer: Optimisation of Critical Parameters Using Physico–Chemical Methods

**DOI:** 10.3390/polym16060764

**Published:** 2024-03-11

**Authors:** Léopold Stampfer, Cécile Bouilhac, Tiphaine Mérian, France Chabert, Toufik Djilali, Valérie Nassiet, Jean-Pierre Habas

**Affiliations:** 1INPT, Laboratoire Génie de Production, École Nationale d’Ingénieurs de Tarbes, Université de Toulouse, 47 Avenue d’Azereix, B.P. 1629, 65016 Tarbes CEDEX, France; leopold.stampfer@enit.fr (L.S.); tiphaine.merian@enit.fr (T.M.); france.chabert@enit.fr (F.C.); 2ICGM, Institut Charles Gerhardt, University Montpellier, CNRS, ENSCM, 34293 Montpellier, France; cecile.bouilhac@umontpellier.fr; 3Lauak Aerostructures France, Parc d’Activité Cognac 8 Rue Louis Caddau, 64500 Tarbes, France; toufik.djilali@groupe-lauak.com

**Keywords:** hydroxyl terminated PEEK, reactive blend, thermal analyses, thermoset-thermoplastic adhesion

## Abstract

Several researchers have examined the interest in using a thermoplastic to increase thermoset polymers’ shock resistance. However, fewer studies have examined the nature of the mechanisms involved between both kinds of polymers. This was the objective of our work, which was carried out using a gradual approach. First, we describe the synthesis of a poly(ether ether ketone) oligomer (oPEEK) with hydroxyl terminations from the reaction of hydroquinone and 4,4′-difluorobenzophenone in N-methyl-2-pyrrolidone. Then, the main physicochemical properties of this oligomer were determined using different thermal analyses (i.e., differential scanning calorimetry (DSC), thermogravimetric (ATG), and thermomechanical analyses) to isolate its response alone. The chemical characterisation of this compound using conventional analytical chemistry techniques was more complex due to its insolubility. To this end, it was sulfonated, according to a well-known process, to make it soluble and enable nuclear magnetic resonance (NMR) and size exclusion chromatography (SEC) experiments. Additional information about the structural and chemical characteristics of the oligomer and its average molecular weight could thus be obtained. The synthesis of an oligoPEEK with α,ω-hydroxyl end-groups and a molecular weight of around 5070 g/mol was thus confirmed by NMR. This value was in accordance with that determined by SEC analysis. Next, the reaction of oPEEK with an epoxy prepolymer was demonstrated using DSC and dynamic rheometry. To this end, uncured mixtures of epoxy prepolymer (DGEBA) with different proportions of oPEEK (3, 5, 10 and 25%) were prepared and characterised by both techniques. Ultimately, the epoxy-oPEEK mixture was cured with isophorone diamine. Finally, topological analyses were performed by atomic force microscopy (AFM) in tapping mode to investigate the interface quality between the epoxy matrix and the oPEEK particles indirectly. No defects, such as decohesion areas, microvoids, or cracks, were observed between both systems.

## 1. Introduction

The scheduled rarefaction and the rising cost of fossil fuels have incited aircraft manufacturers to lighten planes. Most aerospace subcontractor companies are interested in using composite materials lighter than metallic materials usually employed, such as stainless steel, but also metallic alloys such as Inconel^®^ or titanium. However, only metal seems suitable for some elements working in high-temperature conditions, as found with the engine mast. But 80% of the aeroplane parts operate at 200 °C or below. So, substituting metallic parts with pieces based on composite materials seems possible, provided that their specific physical and chemical properties are convenient for the desired application. The list of criteria in the specification includes mechanical resistance to impact. This property is not trivial because the composite materials used in aeronautics are often made of epoxy resin filled with carbon or glass fibres. Indeed, resilience, defined as the ability of the material to absorb and release energy when deformed elastically, is generally lower for epoxy matrices compared to most metals used in the aircraft industry [1,2]. Then, to extend the possible use of this class of materials to external function, much attention has been given in the last few decades to improving the thermal resistance and mechanical properties of epoxy resins. Different pathways were explored, in particular, to increase their toughness by using rubber additives such as siloxane oligomers [3,4], anti-plasticizers [5,6], or even copolymers [7]. To sum up, the nanostructure of filled epoxy systems made it possible to improve epoxy materials’ mechanical and thermal performances slightly. 

Generally speaking, organic composites based on polymeric matrices can be classified into two main categories. The first one, widely developed in the aircraft industry, uses thermoset resin such as epoxy, as described above. The second composite family uses amorphous or semi-crystalline thermoplastic as a matrix. Both generations are generally opposed. Indeed, compared to thermoplastics, thermoset resins present higher mechanical stiffness, reduced creep, and better thermostability due to the presence of aromatic groups in a chemically bonded network.

Moreover, most thermoset resins are liquid before the ultimate crosslinking step. Then, wetting the reinforcing fabric seems more straightforward than that specific to thermoplastics, which are solid at ambient temperature and characterised by higher viscosity in the molten state. Inversely, thermoplastics usually present higher resistance to impact, which can be explained by reversible entanglements that allow more significant macromolecular motions, contrary to fixed knots characteristic of chemical polymeric networks. In other words, no kind of polymeric matrix is perfect. A compromise solution would be to associate thermoplastic and thermoset matrices to combine their properties and produce composite materials with optimised properties. This present study was undertaken within this scientific framework. In such an approach, a significant challenge is the compatibilisation of the thermoplastic with the thermoset polymer to avoid phase separation, which generally occurs with two materials with distinct chemical structures. A possible alternative favours a chemical reaction between thermoplastic and thermoset chains. In other words, the thermoplastic and thermoset macromolecules are covalently bonded together. Such an approach has been tested with epoxy/polyethersulfone blends [8]. To this end, the polyethersulfone (PESU) has been first modified with hydroxyl end groups to make its reaction with epoxy possible. It is now established that PESU use in epoxy formulation improves the toughness of the initial thermoset polymer without a noticeable reduction in its tensile and bending moduli. Some other work has already been conducted on adding modified rubber to an epoxy resin [9]. Such an approach induced a reduction in both crosslinking density and glass transition temperature (Tg). In recent years, much work has been devoted to using functionalised polyaromatic thermoplastics to link them through covalent bonds with thermosetting resins [10,11,12,13,14,15]. All these studies show a synergy between the physicochemical properties of each polymer. In particular, an increase in resilience is usually described [16,17,18,19]. However, the glass transition temperature of the blends remains too low for most of the desired applications. Another approach is described by Francis et al. [20,21]. Their work involved synthesising a hydroxyl-terminated PEEK oligomer to use the OH groups to create covalent bonds with the epoxy resin. The miscibility between the thermoplastic and thermosetting part is increased, and the toughness fracture K_IC_ is improved by about 120% compared to the neat epoxy resin. In comparison, the Tg value is only reduced by 8%. However, in this study, the reaction between hydroxyl-terminated PEEK oligomer and epoxy resin has not been investigated, let alone proven through scientific experiments. The possible cross-linking reaction between the epoxy-grafted PEEK oligomer and the DDS hardener has not been considered. In other words, the PEEK/epoxy miscibility and the interaction phenomena are still poorly understood. 

Our work proposes to complete Francis’s research by investigating the chemical reaction between an oPEEK oligomer’s hydroxyl groups and an epoxy prepolymer’s reactive groups. Firstly, we report the synthesis of a hydroxyl-terminated PEEK oligomer (oPEEK). Its physicochemical properties could be evaluated using different complementary techniques. However, due to its insolubility, its chemical characteristics could not be analysed, and the oPEEK has been sulfonated to allow further chemical study. Then, the chemical reaction between oPEEK and epoxy prepolymer was studied using DSC and thermomechanical analyses. Special attention was paid to the parameters that influence the response. Finally, DGEBA-oPEEK adhesion was investigated and quantified on a sample of the epoxy-oPEEK mixture cured with isophorone diamine by atomic force microscopy (AFM). Such a range of actions, including the demonstration of the reaction between the oligomer and the epoxy resin, is in itself new, given the research already published on this subject.

## 2. Materials and Methods

### 2.1. Chemicals

High-purity hydroquinone (HQ, purity > 99%) and 4,4′-difluorobenzophenone (DFBP, purity > 99%) were purchased from TCI Europe N.V (Paris, France), and anhydrous *N*-methyl-2-pyrrolidone (NMP), toluene (anhydrous), and potassium carbonate (purity > 99%) were procured from Sigma-Aldrich (Darmstadt, Germany). These chemicals were used for the synthesis of oPEEK, which is described hereafter. HQ and DFBP were dried for 12 h at 70 °C in a vacuum chamber, while a higher temperature (120 °C) was preferred to a completely dry potassium carbonate. Then, they were kept in a Schlenk under a nitrogen atmosphere before use. A commercial PEEK (grade 90)—a generous gift from Victrex^®^ company (Lancashire, UK)—was used for comparison. According to the furnisher’s datasheet, this polymer, named PEEK 90 in our study, is characterised by an average molecular weight M_w_ and an average molecular weight M_n_ of 59,600 and 22,300 g/mol, respectively. Concentrated sulfuric acid (96%) was used for the sulfonation of oPEEK for chemical analyses. The resulting polymer has been named soPEEK in this work. Diglycidyl ether of bisphenol-A (DGEBA) epoxy prepolymer with a molecular weight of 340.41 g/mol (trademark DGEBA^®^) was purchased from Sigma-Aldrich (Darmstadt, Germany) under the trademark DER332^®^ and used as received. This grade is thermally stable until 250 °C when subjected to a heating rate of 3 °C/min in a nitrogen atmosphere (see Appendix A in Appendix A). 5-amino-1,3,3-trimethylcyclohexanemethylamine, also known as isophorone diamine (IPDA) from Sigma-Aldrich (Darmstadt, Germany), was used as a hardener with a 170.12 g/mol molecular weight. The respective chemical structures of the epoxy prepolymer and hardener are given in Figure 1.

### 2.2. Blend Preparation

To observe the evolution of the DGEBA-oPEEK reagent mixture by DSC analysis and rheometry, the epoxy prepolymer and PEEK oligomer were mixed with a mechanical stirrer at room temperature for 3 min. To demonstrate DGEBA-oPEEK adhesion by AFM on the blend, the epoxy resin was poured into a beaker set up in a thermoregulated oil bath preheated at 180 °C, according to the work performed by B. Francis et al. [21]. For the first step, 3 wt% oligomer powder was added while maintaining constant mechanical stirring at 300 rpm for 1 h after temperature stabilisation. The oPEEK and DGEBA blend was cooled to room temperature. Then, a stoichiometric amount of IPDA hardener concerning the oxirane functions was added and stirred for a few minutes to obtain a homogeneous mixture. Air bubbles were removed under a vacuum in an oven at ambient temperature for 2 min. Then, rectangular samples were cured for 1 h at 140 °C and 7 h at 190 °C [22].

### 2.3. Fourier Transform InfraRed (FTIR) Analyses

The exact nature of the compounds formed at the different chemical reaction stages was investigated using a Fourier Transform Infrared (FTIR) Spectrum One spectrometer from Perkin Elmer (Waltham, MA, USA). This apparatus was equipped with a Universal Attenuated Total Reflectance (ATR) accessory that made it possible to register the chemical absorption spectra of samples by direct contact on the instrument Zn-Se crystal surface. The experiments were performed in the range 4000–650 cm^−1^ with a resolution of 2 cm^−1^ by accumulating 32 scans. The spectra were interpreted using a well-known spectral library [23].

### 2.4. Nuclear Magnetic Resonance (NMR) Analyses

Nuclear magnetic resonance (NMR) spectra were recorded on a Bruker Avance spectrometer (Billerica, MA, USA) operating at 400 MHz (^1^H) and at room temperature in deuterated DMSO (DMSO-d_6_) as the solvent. The chemical shifts were in parts per million (ppm), where (s) means singlet, (d) a doublet, (dd) a doublet of doublet, (m) a multiplet, and (br) a broad signal. Chemical shifts (^1^HNMR) were referenced to the peak of residual DMSO at *δ* = 2.50 ppm. 

### 2.5. Size Exclusion Chromatography (SEC) Experiments

The accurate characterisation of molar-mass distributions of soPEEK (and indirectly of oPEEK) was carried out by size-exclusion chromatography (SEC). *N*,*N*-dimethylacetamide (DMAc) (with 0.1 wt% LiCl) was used as the eluent at a flow rate of 0.8 mL/min. SEC experiments were performed on a PL-GPC 50 Plus equipped with a Varian model 410 autosampler (Palo Alto, CA, USA). Poly(methyl methacrylate) (PMMA) standard was used for calibration. The SEC apparatus comprised a refractive index detector and was fitted with an 8 μm PolarGel-M pre-column (7.5 × 50 mm) and two 8 μm PolarGel-M columns (7.5 × 300 mm) thermostated at 50 °C. The typical sample concentration was 10 mg/mL.

### 2.6. Differential Scanning Calorimetry (DSC) Analyses

Differential scanning calorimetric (DSC) experiments were performed using a Q200 differential scanning calorimeter from TA Instruments (New Castle, TX, USA) in dynamic mode under a nitrogen flow rate of 50 mL/min. These DSC analyses were first performed on the synthesised oPEEK and the commercial PEEK (90P). Next, the potential reactions between the epoxy/oPEEK prepolymer were examined in the temperature range from −50 °C to 250 °C, with a heating rate of 10 °C/min and for different proportions of oligomer in the mixture.

### 2.7. Thermogravimetric Analyses (TGA)

The thermal stabilities of the oPEEK and commercial PEEK were compared using a TGA2 thermogravimetric analyser from Mettler Toledo (Greifensee, Switzerland). The specimens were set directly on an alumina support plate under a 50 mL/min nitrogen flow rate. The heating ramp was set at 3 °C/min from 25 °C to 800 °C.

### 2.8. Thermomechanical Tests

PEEK 90 and oPEEK were moulded under parallelepiped samples (45 mm × 10 mm × 1 mm) for further thermomechanical analyses using a hydraulic heating press. The manufacturing conditions were defined using the data from the DSC and TGA experiments of these compounds, in particular, to identify the temperature at which the chemical species are completely melted but without any thermal degradation (i.e., 370 °C for PEEK and 350 °C for oPEEK for 10 min). The thermomechanical properties of oPEEK and PEEK 90 were evaluated using a stress-controlled dynamic rheometer AR2000Ex from TA Instruments (New Castle, TX, USA) equipped with rectangular torsion geometry. These rheological tests were conducted at a 5 °C/min heating rate from −150 °C to 180 °C. The applied strain was set at 0.1% to check the linear viscoelastic domain while the shearing angular frequency was kept constant (ω = 1 rad/s). This latter value was low enough to allow the temperature Tα, taken at the maximum of the α peak on G″, to be considered a reliable evaluation of the cured polymer’s glass transition temperature (Tg).

To monitor the chemical reaction between the epoxy prepolymer and the oPEEK oligomer, thermomechanical tests were performed on an ARES rheometer from Rheometrics/TA Instruments (New Castle, TX, USA) under airflow, using a 25 mm diameter parallel-plate configuration in dynamic mode. Once more, all the tests were carried out in the viscoelastic linear domain, which was previously determined from dynamic strain sweep analyses.

### 2.9. Atomic Force Microsccopy (AFM)

The surface characterisation of cured IPDA/DGEBA samples was performed by atomic force microscopy (AFM) using an Asylum MFP-3D Infinity (Santa Barbara, CA, USA). A silicon tip (AC240TS-R3 model) with a stiffness of 2 N/m and a resonant frequency of 70 kHz was used. The experiments used the “tapping” mode for topographic analysis. The surface sample thickness of 100 µm was obtained using an ultra-microtome LEICA EM UC7 (Wetzlar, Germany) working at ambient temperature.

### 2.10. Synthesis of oPEEK Oligomers 

oPEEK synthesis is presented according to a pathway already described in the literature but adapted to our reactants [21]. It is a bi-step procedure that is roughly summarised as based on the nucleophilic substitution reaction of DFBP with HQ (Figure 2). The desired value for the average number molecular weight (M_n, targeted_ = 5700 g/mol) of this oligomer was controlled using modified Carothers’s equation. The presence of OH endcaps was ensured by taking a slight excess of HQ.

A four-necked round bottom flask was equipped with a mechanical stirrer and a Dean–Stark trap outfitted with a condenser to proceed with the synthesis. The flask was carefully dried and purged with dry nitrogen before introducing the chemicals. First, 40 g (0.24 mol) of TBHQ and 39.9 g (0.29 mol) of potassium carbonate were poured into a mixture of NMP (340 mL) and toluene (150 mL). Then, the reaction medium was heated up to 175 °C in an inert atmosphere (N_2_) under reflux and continuous mechanical stirring (100 rpm) for 24 h. Water produced during this reaction was purged from the medium as an azeotrope with toluene through the Dean–Stark equipment. The volume of water removed and FTIR analyses registered on regular sampling in the reaction medium were used to evaluate the reaction kinetics and ensure the total consumption of the reactive species. After 24 h, the reaction temperature was reduced to 100 °C. Then, 48.2 g (0.22 mol) of DFBP and further NMP (100 mL) were added to the synthesis flask. Then, the mixture temperature was increased to 175 °C, and mechanical stirring was continued for 24 h with regular sampling for chemical analyses. Finally, the reaction mixture was cooled to room temperature and poured into a large volume of distilled water. The solid precipitate was filtered and washed in a significant excess of methanol under magnetic stirring for 7 h. After filtering, the brown solid substrate was dried in a temperature-controlled vacuum chamber set at T = 110 °C.

M_n, NMR_ = 5070 g/mol deduced from M_n, NMR_ of soPEEK.FTIR (cm^−1^): 3045 ((CH)=CH), 1642 (-C=O), 1595 (-C=C- aromatic), 1491 (-C=C- aromatic), 1224 (-C-O-C), 1190 (-C-O-C-), 1149 (-C-O-C-), 1110 (C-O-C), 837 (C-H aromatic).

### 2.11. Sulfonation of oPEEK (Synthesis of soPEEK)

It has been shown in the literature that a PEEK with a high sulfonation degree (SD) presents reduced crystallinity compared to the initial polymer because of the presence of grafted sulfonic acid (-SO_3_H) groups that disrupt the macromolecular linearity [24,25]. Indeed, the primary interaction between the amorphous macromolecular chains is based on electrostatic forces between the polar sulfonic acid groups. A sulfonation degree (SD) higher than 40% is usually necessary to enable the solubilisation of PEEK in polar solvents such as dimethylformamide (DMF), dimethylacetamide (DMAc), and dimethylsulfoxide (DMSO) at room temperature, allowing its characterisation by liquid NMR and SEC. Then, this chemical pathway was retained in our work to permit the further study of oPEEK using chromatography working with liquids. Note that the efficiency of this approach to characterise PEEK by size exclusion chromatography was confirmed in the literature [26]. The sulfonation step of the oPEEK oligomer was performed at 50 °C for 6 h with 96% sulfuric acid. It was operated by dissolving 5 g of oPEEK into 200 mL of sulfuric acid (96%) previously set in a bi-necked round bottom flask under an inert atmosphere (Figure 3). The reaction medium was heated to 50 °C under vigorous mechanical stirring (700 rpm) for 6 h. Then, the solution was poured into a bath of cold distilled water. The precipitate isolated by filtration was washed in pure water twice. Finally, after new filtration, the dark brown substrate (soPEEK) was dried for 5 h in a vacuum chamber set at T = 120 °C. 

^1^H NMR (400 MHz, DMSO-d_6_) δ (ppm) presented in Appendix A (Appendix A): 7.86–7.69 (m, -C_6_**H**_4_(C=O)C_6_**H**_4_-, H^A^+H^A’^; 65H), 7.52 (s, -O-C_6_**H**_3_(SO_3_H)-O-, H^E^, 9H), 7,27 (s, -O-C_6_**H_4_**-O-, H^C’^, 30H), 7.26–7.20 (dd, -O-C_6_**H**_3_(SO_3_H)-O-, H^C^, 9H), 7.18 (m, -C_6_H_4_(C=O)C_6_**H**_4_-, H^B^, 34H), 7.17–7.12 (m, -O-C_6_**H**_3_(SO_3_H)-O-, H^D^, 9H), 7.09 (m, -O-C_6_**H**_3_(SO_3_H)-OH, H^C2^, 2H), 7.01 (d, -C_6_**H**_4_(C=O)C_6_H_4_-, H^B’^, 34H), 6.89 (d, -O-C_6_**H**_3_(SO_3_H)-OH, H^C1^, 2H), 5.67 (br, -O**H**).M_n, NMR_ = 5900 g/mol.SEC (DMAc, RI detector, PMMA calibration): M_n_ = 7900 g/mol, *Đ* = 1.60.FTIR (cm^−1^) presented in Appendix A (Appendix A): 2821 (OH), 1644 (-C=O), 1583 (-C=C- aromatic), 1492 (-C=C- aromatic),1280 (O=S=O), 1223 (-C-O-C-), 1149 (-C-O-C-), 1110 (C-O-C), 1074 (O=S=O), 1009 (O=S=O), 837 (C-H aromatic), 704 (S-O). 

## 3. Results

### 3.1. Determination of Chemical Characteristics

#### 3.1.1. FTIR Analyses of oPEEK Oligomer and PEEK Polymer

Figure 1 shows the FTIR spectrum of oPEEK and that of the commercial PEEK 90P for comparison. As expected, both spectra are similar and present absorption bands characteristic of the same chemical structure. No discrimination is possible between an oligomer and polymer FTIR spectrum. Both spectra show an absorption band in the 1450–1500 cm^−1^ range corresponding to C=C stretching vibrations of the aromatic ring. The band in the 1220–1225 cm^−1^ range is attributed to the ether bond (C-O-C). The carbonyl (C=O) stretching vibration was observed in the 1600–1670 cm^−1^ range. In the oPEEK spectrum, no broadband characteristic of hydroxyl groups is visible between 3000 and 3500 cm^−1^. This result is likely due to a possible “dilution” effect induced by the length of the central chemical skeleton. In other words, FTIR analysis cannot conclusively establish the presence of the desired hydroxyl end groups on the oPEEK.

#### 3.1.2. Physico–Chemical Analyses of soPEEK

As currently observed with PEEK, the oPEEK oligomer was insoluble in almost all solvents classically used in analytical chemistry (chloroform, DMSO…) except for some strong acids such as concentrated sulfuric acid. This insolubility made its chemical characterisation by liquid NMR and SEC impossible. However, it is clearly shown in the literature that the solubility of this kind of polymer can be significantly improved after its sulfonation [24,27,28]. 

The FTIR spectrum of soPEEK is drawn in the Appendix A as Appendix A. It is quite different from that characteristic of the original compound, i.e., before the sulfonation process. Indeed, several absorbance bands are specific to the sulfonic groups. For instance, those centred at 1009, 1074, and 1280 cm^−1^ are characteristic of symmetric and asymmetric stretching vibrations of sulfonic acid groups [29]. The band initially located at 1490 cm^−1^ in the oPEEK spectrum that is representative of the aromatic C-C bond is observed to split into two distinct peaks in the soPEEK spectrum (resp. 1476 and 1493 cm^−1^) due to the grafting of sulfonated groups. 

As expected, the soPEEK obtained was soluble at ambient temperature in DMAc and DMSO. Thus, its characterisation by ^1^H NMR spectroscopy could be achieved. Appendix A presents the corresponding data and their interpretation in the Appendix A. Given the experimental conditions and especially the stoichiometry of the reactants, the polymer is likely to be end-capped with a hydroquinone ring on both sides. Then, the average molecular weight of sulfonated oPEEK was evaluated to be close to 5900 g/mol. Thus, it becomes possible to estimate the molecular weight of the starting oPEEK to be close to 5070 g/mol. This value agrees with the targeted molecular weight based on Carothers’s equation, which was about 5700 g/mol. In other words, it shows that the oPEEK with the wished molecular weight was finally obtained. 

A Size Exclusion Chromatography experiment (SEC) was performed on the soPEEK to evaluate its molecular size further. The analysis was carried out using DMAc as the eluent, and the corresponding data are presented in Figure 2. The curve of the intensity registered as a function of elution time shows the presence of a broad mass, which seems to be made of three elementary peaks.

After calibration using PMMA as a reference, each peak could be attributed to the presence of oligomers characterised by the molecular weights Mn of 3700, 7900, and 20,000 g/mol, respectively. Due to the predominance of the central peak, the average value M_n_ of the soPEEK is about 7900 g/mol. This value is slightly higher than that evaluated by the NMR experiment (5070 g/mol).

New evidence about the oligomeric nature of the oPEEK could be established using thermal analyses.

### 3.2. Thermal Analyses

#### 3.2.1. DSC Characterisation of oPEEK and PEEK 90

DSC thermograms of synthesised oPEEK and commercial PEEK are shown in Figure 3. Considering first the thermal behaviour of PEEK, the discontinuity centred at around 150 °C is characteristic of the glass transition temperature of the polymer amorphous phase (Tg_(PEEK)_ ≈ 150 °C). Next, the melting of the crystalline phase is observed by the presence of a single narrow endothermic peak beginning at 298 °C. In the case of the oPEEK, this latter zone produces itself at lower temperatures. Indeed, the inception of the endothermic area is measured close to 245 °C. This reduced melting temperature supports that the oPEEK presents a smaller average molecular weight than that characteristic of the PEEK polymer. In the case of the oligomer, the thermal melting zone seems to be made up of two individual peaks that are likely characteristic of the melting of crystallites with distinct populations and morphologies. Such behaviour has already been reported in the literature by Bas et al. [30] and Jonas et al. [31].

The melting enthalpy ΔH_m_ during the heating is used to calculate the degree of crystallinity from Equation (1). The melting enthalpy $\Delta H_{m}^{100\%}$ of an ideal crystal of PEEK is 130 J/g [32], and $W_{\mathrm{polymer}}$ is the weight fraction of the polymer matrix, which is equal to 1 in our case.
(1)$${𝒳}_{C}=\frac{\Delta H_{m}}{\Delta H_{m}^{100\%}W_{\mathrm{polymer}}}\;\times \;100$$

The melting enthalpy is about 92 ± 2 J/g for the oPEEK and 45 ± 3 J/g for the PEEK 90, respectively. The crystallinity fraction calculated from (1) is 71 ± 1% for the oPEEK and 35 ± 1% for the PEEK 90, respectively. The more substantial crystallinity rate of oPEEK is another element that supports its low molecular weight and may explain why oPEEK glass transition temperature is not evident in Figure 3.

#### 3.2.2. Thermogravimetric Analyses of oPEEK and PEEK 90

A TGA experiment evaluated both compounds’ thermal stability under nitrogen. Figure 4 presents the relative residual weight as a function of the temperature for synthesised oPEEK and commercial PEEK 90 for comparison. The thermogravimetric behaviour registered with PEEK polymer agrees with that described previously in the literature [32].

The weight loss of 2% observed for oPEEK between 100 °C and 150 °C is attributed to the evaporation of absorbed water, which is likely to occur because of hydroxyl end groups in the oligomer structure. The degradation onset registered at the drop of the TGA curve is much reduced than that of PEEK 90 (496 ± 3 °C against 545 ± 3 °C, respectively). The lower thermostability of the oPEEK is likely a consequence of the smaller size of its constituting molecular chains. This result supports that oPEEK is an oligomer, not a polymer, contrary to PEEK 90. 

#### 3.2.3. Thermomechanical Analyses of oPEEK and PEEK 90

To further the comparative study of oPEEK and PEEK, the viscoelastic properties of these compounds were measured as a function of temperature in dynamic mode (ω = 1 rad/s). Figure 5 presents the data registered between 20 and 340 °C with a heating rate of 3 °C/min.

Not surprisingly, the thermomechanical profile of PEEK 90 corresponds to that of a thermoplastic with a semi-crystalline morphology [33]. Indeed, at low temperatures, a first domain characterised by the predominance of the storage modulus G′ (~1 GPa) on the loss modulus G″ can be observed between 20 °C and 125 °C. In this domain, the decrease of both moduli with temperature remains somewhat limited. Then, the presence of a peak on the G″ curve and an inflexion on the G′ curve are simultaneously detected. Both phenomena are characteristic of the main mechanical relaxation of the polymer amorphous phase, which is associated with the glass transition phenomenon. The temperature, named T_α_, taken at the maximum of G″ peak, yields a reliable evaluation of the T_g_ of the polymer since the experiment was conducted at the pulsation ω = 1 rad/s. It is measured as being close to 150 °C. Above T_g_, the crystalline phase remains almost immobile, which allows the complex modulus to be maintained at high values, defining the so-called “crystalline plateau”. Finally, a rapid drop in both moduli is representative of crystallite fusion. The inception of the melting area T_m_ is thus detected close to T = 300 °C. It should be noted that these critical temperatures agree with those measured directly by DSC.

The thermomechanical profile of oPEEK also consists of different thermal domains, which prove its semi-crystalline morphology. However, significant differences are also observed in comparison with the PEEK polymer. The mechanical stiffness of the oligomer, its glass transition temperature (120 °C), and its melting temperature (275 °C) are significantly lower than those previously measured with PEEK 90. This difference, which agrees with the DSC data, can be explained by the difference in molecular size between the oligomer and the polymer. The oPEEK relaxation peak’s lower amplitude compared to that of PEEK reveals that the oligomer has a much-reduced amorphous phase than the polymer. In other words, oPEEK is more crystalline than PEEK 90, which is consistent with the findings of the thermal analyses carried out previously.

### 3.3. Study of the Reaction between DGEBA and oPEEK

As explained in the introductory part, oPEEK was synthesised to study its possible reaction with DGEBA for further improvement of the mechanical properties of the thermoset resin. The corresponding scheme involves the reaction between the OH endcaps of oPEEK with the oxirane groups of the epoxy prepolymer DGEBA via nucleophilic addition reaction. This reaction simultaneously induces the formation of secondary hydroxyl groups and ether links (Figure 4). A significant excess of the epoxy prepolymer was used to ensure that oxirane groups formed both terminations for further reaction with the diamine hardener. For instance, in our study, a blend with 25% by weight of oPEEK (having an estimated M_n_ value equal to 5070 g/mol) and 75% by weight of epoxy prepolymer (M_n_ = 340.41 g/mol) corresponds to a blend with about 45 times more DGEBA molecules than oligomer chains.

#### 3.3.1. Characterisation of DGEBA-oPEEK Blend (75–25 wt%)

FTIR analysis of oPEEK showed that detecting the presence of terminal OH groups was impossible. Consequently, for the same reason, this technique is not convenient to investigate the possible reaction between the oxirane groups of DGEBA and the hydroxyl groups that endcap the oPEEK (Figure 4). Then, an alternative experimental approach had to be considered to examine the above-mentioned reaction. Temperature-dependent DSC experiments were conducted under a nitrogen atmosphere on freshly prepared DGEBA-oPEEK mixtures in different proportions. 

Figure 6 presents the results registered with a DGEBA-oPEEK mixture with 75% and 25% weight fractions for successive runs between −50 °C and 250 °C. During the first heating, the initial glass transition of the mixture can be identified by the presence of a discontinuity in the thermogram. Its value is evaluated to be close to −21 °C and corresponds to the epoxy prepolymer glass transition temperature (T_g0_). Then, as the temperature increases, a wide exothermic signal is detected between 100 °C and 250 °C in the DSC thermogram. This exotherm seems to be made by two peaks centred at 150 °C and 225 °C, respectively, and is even preceded by a smaller peak centred at 90 °C. This whole thermal zone is interpreted as characteristic of the reaction between the epoxy groups of DGEBA with the OH terminations of oPEEK. Different elements support this hypothesis. The first one is the slight increase in the T_g_ of the mixture (Δ = +5 °C) measured during the second run, which can be explained as the result of the formation of macromolecules with higher size (i.e., epoxy terminated PEEK). The second element is the quasi-disappearance of the exothermic zone initially detected in the first run. This indicates that the DGEBA-oPEEK reaction was achieved, and logically, no further evolution is observed for consecutive analyses. 

A last element helps us understand that the DGEBA-oPEEK reaction produces itself according to different steps. The thermomechanical profile of oPEEK detailed above shows that the oligomeric units have relatively low molecular mobility at temperatures below 100 °C. Then, between 0 °C and 100 °C, the only possible reaction between DGEBA and oPEEK is the OH terminations that are directly accessible to the epoxy terminations. This reaction is somewhat limited and is detected by different exothermic peaks with reduced amplitudes. At higher temperatures, especially within the glass transition range of oPEEK between 100 and 150 °C, the increased molecular mobility of the amorphous phase allows the OH-epoxy reaction to resume to a greater extent. The response becomes even more intense as the temperature approaches the thermal range associated with the melting of the crystalline regions of oPEEK (T ≅ 240 °C). Indeed, the oligomer chains can diffuse more efficiently; therefore, the previous reaction is more likely to occur. This last phase of thermal reactivity is responsible for forming the third peak in the DSC thermogram.

By inducing the formation of larger molecular chains, the reaction between oPEEK and DGEBA should logically also be monitored by conducting rheological measurements. To this end, a fresh DGEBA-oPEEK mixture with unchanged proportions was analysed by dynamic rheometry (5 °C/min) as a function of temperature. The corresponding data are plotted in Figure 7.

At low temperatures (i.e., 50 °C), the mixture is characterised by low values of G′ and G″ (about 200 Pa), and the viscous character is predominant (G″ > G′). Then, above T = 42 °C, the respective values of both moduli increase significantly with temperature. The elastic contribution becomes predominant for T > 60 °C (G′ > G″). A new increase is observed for T > 80 °C, and both G′ and G″ curves show a broad peak centred at T = 150 °C. A further increase in moduli is observed when the temperature exceeds 175 °C and seems to reach a limit for T = 250 °C. Given the interpretation of the calorimetric behaviour of this blend detailed before, this evolution can be explained as being the consequence of the multi-step reaction of the hydroxyl terminations of oPEEK with the epoxy groups of DGEBA. Indeed, different specific thermal zones can be found in agreement with those detailed by the DSC analysis. For T < 90 °C, the moduli values decrease with temperature, reminiscent of the behaviour of conventional fluids, showing that no reaction occurs. Above 90 °C, the slight increase in the viscoelastic moduli is due to the OH-epoxy reaction, but this concerns quasi-immobile oPEEK chains, since this critical temperature is lower than the T_g_ of the oligomer. In the glass transition zone of oPEEK, i.e., between 100 and 150 °C, the increased molecular mobility of the amorphous phase allows the OH-epoxy reaction to occur to a greater extent. Finally, the second peak observed in the rheological curves for T > 175 °C is interpreted as characteristic of a faster and higher reaction due to the proximity of the melting zone.

#### 3.3.2. Calorimetric Study of DGEBA-oPEEK Reactive Blends with Different Proportions

To go further in the calorimetric study of the DGEBA-oPEEK reaction, a new series of blends was prepared with 3, 5, and 10% by weight of oPEEK particles for subsequent analysis by DSC. Given the specific molecular weights M_n_ of oPEEK (5070 g/mol) and DGEBA (340 g/mol), these mixtures comprise approximately 482, 283, or 134 times more DGEBA molecules than oPEEK oligomer chains, respectively. Figure 8 shows the corresponding DSC thermograms, performed under the same experimental conditions as those used previously for the DSC analysis of the mixture based on 25% by weight of oPEEK, which is recalled for more accessible discussion. For all blends, the heat flux was reduced to the number of OH moles to quantify the proportions of epoxy groups involved in the reaction. 

The area of the whole exothermic zone attributed to the occurrence of the DGEBA-oPEEK reaction was integrated to evaluate the corresponding enthalpy value normalised by the moles of OH groups present in oPEEK (Table 1). Then, it can be noted that the enthalpy decreases as the oPEEK content increases.

This phenomenon can be understood by assuming that at higher oPEEK proportions, oPEEK grains form aggregates. Given that the OH-epoxy reaction only involves hydroxyl groups present at the surface of oPEEK grains, a higher proportion of oligomer will induce a higher reaction yield if the grains are dispersed individually in the DGEBA liquid.

Based on these results, the final part of our research aimed to investigate the production of rigid materials based on DGEBA previously treated with 3 wt% oPEEK and then cured with the IPDA hardener.

### 3.4. Morphological Analyses by AFM

Using the previous conclusions, DGEBA-oPEEK 3% was mixed and cured with IPDA according to the methodology detailed in the “blend preparation” section. Atomic Force Microscopy analyses were performed in tapping mode to obtain topological characterisations of a 20 × 20 µm^2^ polymeric surface. Figure 9 shows both topography (a) and phase results (b) registered with a polymeric surface made with epoxy matrix (greyscale) and oPEEK inclusion (white mark). In other words, each picture shows, at its centre, the presence of the same oPEEK cluster surrounded by the cured epoxy matrix.

A red line indicates the horizontal displacement of the probe along the surface. It first runs along the surface of the epoxy matrix, then passes over the oPEEK inclusion before ending again with the characterisation of the epoxy surface. The results of the topographical analysis presented in Figure 9a enable us to study the precise profile of the surface. This characterisation has been performed on three zones for two samples. The root mean square roughness (RMS) measured on an area of epoxy surface is 3.2 ± 0.2 nm, which proves the good quality of the ultramicrotome preparation. Consequently, the traces induced by the cutting of the sample by the microtome knife are detected by the presence of various peaks that are visible on the part of the topological picture without oPEEK particle. Then, a larger mass is observed. This is characteristic of the oPEEK inclusion with a maximum roughness of 8 nm. No space between the fillers and the matrix is detected, indicating the absence of decohesion area, microvoids, or cracks between the epoxy matrix and the oPEEK cluster. Figure 9b presents the data of the phase evolution registered during the same scan of the AFM probe. It shows a higher phase in the zone characteristic of the oligomer compared to the zone specific to the cured epoxy matrix. This result can be explained by the fact that the oligomer comprises individual molecular chains. In contrast, the epoxy matrix is based on a three-dimensional polymer network, which is intrinsically less dissipative. Once more, no anomaly is observed at the interface between the oPEEK filler and the epoxy matrix, confirming the topological analyses.

## 4. Conclusions

The interest in using a thermoplastic polymer or oligomer in a thermoset matrix is often described in the literature, through mechanical tests. Nevertheless, to our knowledge, little evidence has been proposed up to now about the nature of the interaction between both kinds of materials. In this paper, we wish to make a worthy contribution by studying, in particular, the combination of a laboratory-made hydroxyl-terminated PEEK oligomer with an epoxy matrix. Several elements were clearly shown through a gradual scientific approach. First, the presence of hydroxyl endcaps and the average molecular weight of the oligomer were determined via the study of its sulfonated counterpart. However, the small proportion of OH groups per oligomer oPEEK (i.e., 2) did not make the observation of their reaction with epoxy groups by analytical chemistry possible.

Nevertheless, a physicochemical methodology based on the combination of DSC and thermomechanical experiments proved effective in demonstrating this reaction occurrence. These analyses revealed that the reaction occurs in several steps. Each step takes place within a specific thermal range characteristic of different molecular mobility states of the oPEEK units. The best reaction yield was observed with a weight proportion of oPEEK close to 3%. This result indicated the possible formation of oligomer clusters at higher oPEEK ratios. The interface quality between the thermoplastic oligomer and the modified and cured DGEBA was good based on AFM experiments. Considering all these findings, various perspectives will be investigated in future work.

The first area of work will be to study the behaviour of the oPEEK-DGEBA reactive mixture with the same proportion of oligomer but different grain sizes. Logically, smaller granulometry should obtain a better reaction yield because they have grains with higher surface/volume ratios. The second area of work will be the optimisation of the dispersion of oPEEK grains. Several approaches (mechanical processing, ultrasonic techniques…) will be compared. Then, after the optimisation of these parameters, the mechanical properties of oPEEK-DGEBA-IPDA blends will be quantified through different experiments (tensile, flexural, and impact tests) and compared with those of materials produced only by crosslinking the epoxy prepolymer with the diamine hardener.

## Data Availability

Data are contained within the article and Appendix A.

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
