# Peer review of "Investigation of the Reaction between a Homemade PEEK Oligomer and an Epoxy Prepolymer: Optimisation of Critical Parameters Using Physico–Chemical Methods"

_polymers, 2024, doi:10.3390/polym16060764_

Round 1

Reviewer 1 Report

Comments and Suggestions for Authors

Polymers

This manuscript deals with the synthesis of a poly(ether ether ketone) oligomer (oPEEK) with hydroxyl terminations from the reaction of hydroquinone and 4,4’-difluorobenzophenone in N-methyl-2-pyrrolidone. However, this manuscript need to further emphasize the innovation of this work. Which is the main objective of this work? Where is a innovative point of this work? Which characterization provide different techniques use in this work?  Thus some problems should be addressed before consideration for publishing. Additionally, English should be improve.

11   Abstract should be rewritten, the main achievement of this work should be stress in the Abstract. Address all main achievement of this work obtained from the results. For instance, the sentences:

“The main physicochemical properties of the oligomer could be determined using different thermal analyses. But, as this compound was insoluble, it was sulfonated to make its consecutive chemical characterization possible through NMR and SEC experiments. “

aa)     This is really achievement of the work? If yes, please compare with other study published previously by other researchers?

bb)     This is first time that authors use sulfonated compound? The idea of sulfonated come out form the authors o this is achievement of other researchers?

Cc)     Authors indicated chemical characterization by NMR and SEC. Which kind of chemical properties you can extract form SEC? Is chemical or physical characteristic? Please revise whole manuscript from this point of view. Addressed well properties which can be extracted from each technique.

dd)     Please identify which technique you use for thermal analysis and use whole technique names and not only abbreviation. Nuclear magnetic resonance (NMR), etc.

22     Which is the aim and innovation of this work if compare with state of art? Addressed this in the end of Introduction

33     Please clearly defined the origin of all substances specifying the producing company, country and city. The same for the equipment used for characterization.

44     Please rewritten the sentence “Finally, DGEBA-oPEEK adhesion was investigated and quantified on a sample of the epoxy-oPEEK mixture cured with isophorone diamine by AFM” What you quantified by AFM? How you were able to quantify this parameter? Why you investigated only one sample? Are you make only one AFM measurement for this system?

55      How you calculate the roughness by AFM when you have some meny lines related to the cutting affected surface and roughness measument?

66      In technical manuscript “hour” should be “h”. Please revised whole manuscript for this point.

77      You conclude good adhesion by AFM techniques? Simple AFM measurement is not quantitative technique. You see good adhesion in comparison with which system? How you are able to conclude good adhesion for single system? Please revise this and show related references, where other authors show similar results.

88      How you prepare the sample for AFM? Thermosetting samples need special preparation to see the morphology. Please identify this point in your manuscript. Are you sure that your thickness was around 100 micrometer? This is valid preparation for AFM? Which is the roughness limit which can be scan by your AFM?

99     Why you study your materials using different heating rates ranging from 6 to 12°C/min? Where you show these results? Which was the reason to perform different DSC measurements? This is not clear in manuscript. Please specify how you measure (range, velocity) your sample during reaction and after curing and why you use different velocity for monitoring curing reaction.

Comments on the Quality of English Language

Extensive editing of English language required

Reviewer 2 Report

Comments and Suggestions for Authors

The authors proposes to create a composition of a classical thermoset polymer (epoxy resin) with high temperature thermoplastic (PEEK) bearing in mind to improve mechanical properties of such composition, and first of all, its resistance to shocks.

This concept aiming to improve the properties of epoxy resin is not new. However, the proposed using of PEEK and the complete scenario of preparing composition looks rather interesting.

The whole synthetic and analytic parts of the article are not objectionableÑŽ

 However, I do not see the main result of the study: what material you received and what are their mechanical properties. The authors promise to show it later, but the study without this final result losses its value.

Round 2

Reviewer 1 Report

Comments and Suggestions for Authors

Accept in present form.

Author Response

Thank you again for your review of our work. You have greatly helped us to improve the quality of our manuscript.

Reviewer 2 Report

Comments and Suggestions for Authors

The title change (using physico-chemistry) seems to me stylistically somewhat crooked. I would prefer something different, e.g. using physico-chemical methods

Author Response

Thank you very much for your constructive comments on our article. Of course, we accept the last change you proposed, which is once again a good one.